# Influenza A virus resistance to 4'-fluorouridine coincides with viral attenuation *in vitro* and *in vivo*

Carolin M. Lieber[1], Hae-Ji Kang[1], Megha Aggarwal[1], Nicole A. Lieberman[2], Elizabeth B. Sobolik[2], Jeong-Joong Yoon[1], Michael G. Natchus[3], Robert M. Cox[1], Alexander L. Greninger[2], Richard K. Plemper[1]*

1 Center for Translational Antiviral Research, Georgia State University Institute for Biomedical Sciences, Atlanta, Georgia, United States of America, 2 Virology Division, Department of Laboratory Medicine and Pathology, University of Washington Medical Center, Seattle, Washington, United States of America, 3 Emory Institute for Drug Development, Emory University School of Medicine, Atlanta, Georgia, United States of America

* rplemper@gsu.edu

**Data Availability Statement:** All relevant data are within the manuscript and its Supporting Information files.

## Abstract

Pre-existing or rapidly emerging resistance of influenza viruses to approved antivirals makes the development of novel therapeutics to mitigate seasonal influenza and improve preparedness against future influenza pandemics an urgent priority. We have recently identified the chain-terminating broad-spectrum nucleoside analog clinical candidate 4'-fluorouridine (4'-FIU) and demonstrated oral efficacy against seasonal, pandemic, and highly pathogenic avian influenza viruses in the mouse and ferret model. Here, we have resistance-profiled 4'-FIU against a pandemic A/CA/07/2009 (H1N1) (CA09). *In vitro* viral adaptation yielded six independently generated escape lineages with distinct mutations that mediated moderate resistance to 4'-FIU in the genetically controlled background of recombinant CA09 (recCA09). Mutations adhered to three distinct structural clusters that are all predicted to affect the geometry of the active site of the viral RNA-dependent RNA polymerase (RdRP) complex for phosphodiester bond formation. Escape could be achieved through an individual causal mutation, a combination of mutations acting additively, or mutations functioning synergistically. Fitness of all resistant variants was impaired in cell culture, and all were attenuated in the mouse model. Oral 4'-FIU administered at lowest-efficacious (2 mg/kg) or elevated (10 mg/kg) dose overcame moderate resistance when mice were inoculated with 10 LD$_{50}$ units of parental or resistant recCA09, demonstrated by significantly reduced virus load and complete survival. In the ferret model, invasion of the lower respiratory tract by variants representing four adaptation lineages was impaired. Resistant variants were either transmission-incompetent, or spread to untreated sentinels was fully blocked by therapeutic treatment of source animals with 4'-FIU.

**Funding:** This study was supported, in part, by public health service grants AI141222 (to R.K.P.) and AI171403 project 1 (to R.K.P.) from the NIH/NIAID. The funders had no role in study design, data collection and analysis, decision to publish, or preparation of the manuscript.

**Competing interests:** MGN is a coinventor on patent WO 2019/1736002 covering composition of matter and use of 4'-FlU (EIDD-2749) and its analogs as an antiviral treatment. This study could affect his personal financial status. RKP reports contract testing from Enanta Pharmaceuticals and Atea Pharmaceuticals, and research support from Gilead Sciences, outside of the described work. ALG reports contract testing from Abbott, Cepheid, Novavax, Pfizer, Janssen and Hologic, research support from Gilead, outside of the described work. All other authors declare that they have no competing interests.

## Author summary

Reduced sensitivity to FDA-approved influenza drugs is a major obstacle to effective antiviral therapy. We have previously demonstrated oral efficacy of a novel clinical candidate drug, 4'-FlU, against seasonal, pandemic, and highly pathogenic avian influenza viruses. In this study, we have determined possible routes of influenza virus escape from 4'-FlU and addressed whether resistance imposes a viral fitness penalty, affecting pathogenicity or ability to transmit. We identified three distinct clusters of mutations that lead to moderately reduced viral sensitivity to the drug. Testing of resistant variants against two chemically unrelated nucleoside analog inhibitors of influenza virus, conditionally approved favipiravir and the broad-spectrum SARS-CoV-2 drug molnupiravir, revealed cross-resistance of one cluster with favipiravir, whereas no viral escape from molnupiravir was noted. We found that the resistant variants are severely attenuated in mice, impaired in their ability to invade the lower respiratory tract and cause viral pneumonia in ferrets, and transmission-defective or compromised. We could fully mitigate lethal infection of mice with the resistant variants with standard or 5-fold elevated oral dose of 4'-FlU. These results demonstrate that partial CA09 escape from 4'-FlU is feasible in principle, but escape mutation clusters are unlikely to reach clinical significance or persist in circulation.

## Introduction

Seasonal influenza viruses cause a significant public health and socioeconomic burden [1,2]. Approximately one billion people are infected annually worldwide [3], and millions require hospitalization and advanced care. Case fatalities exceed 600,000 in interpandemic years, and can be substantially higher when zoonotic influenza viruses spill over into the human population, causing large-scale pandemics [4]. A multivalent annual influenza vaccine provides moderate protection, but vaccine efficacy is moderate under the best of circumstances, and compromised in particular in the most vulnerable patient populations such as older adults and the immunocompromised [5–7]. Benefit rapidly declines when vaccines are poorly matched with circulating virus strains or when novel, pandemic virus strains emerge [8,9].

Three different classes of antivirals are currently FDA approved for use against influenza: the adamantes (i.e. amantadine), which block the viral M2 channel [10]; the inhibitors of viral neuraminidase (NA) activity (i.e. oseltamivir phosphate) [11,12]; and the PA endonuclease blocker baloxavir marboxil [13,14]. Although originally effective when administered early after infection, each of these classes has a low genetic barrier against viral resistance. Excessive veterinary use of the adamantes has led to widespread preexisting presence of the signature M2 S31N resistance mutation in circulating human and animal influenza A virus (IAV) strains [15,16], and their use for influenza therapy is no longer recommended by the Centers for Disease Control and Prevention since 2006 [17,18].

Whereas neuraminidase inhibitors are still in clinical use, resistance mutations emerge rapidly in the viral NA protein in treatment-experienced virus populations and have been found in circulating strains. For instance, over 99% of seasonal H1N1 isolates were oseltamivir-resistant in April 2009, immediately before the emergence of the 2009 swine origin IAV pandemic [19,20]. Although pandemic 2009 (pdm09) H1N1 strains were initially sensitive to oseltamivir, pre-existing resistance emerged in later stages of the pandemic and high transmission of oseltamivir-resistant pdm09 viruses was detected in some communities [20,21].

The most recent addition to the antiviral arsenal against influenza, baloxavir marboxil, demonstrated strong efficacy after administration of a single dose [22]. However, treatment-

emergent resistance, a signature PA I38T/M/F substitution, appeared almost instantaneously, resulting in rebound of virus replication in 10% of treated patients in a phase 3 clinical trial [14,23].

The nucleoside analog inhibitor favipiravir, which was conditionally approved in Japan for treatment of pandemic influenza virus infections when other influenza drugs are ineffective [24,25], has a substantially higher barrier against viral escape. Favipiravir-resistant variants did not emerge in clinical trials [24], but an *in vitro* adaptation identified a K229R mutation in the PB1 subunit of the RdRP complex that reduced viral susceptibility to the drug when combined with a compensatory PA P653L mutation that partially restored viral fitness [26]. It is uncertain, however, whether escape from favipiravir may amount to a clinical problem.

We have recently identified 4'-FlU, an orally efficacious nucleoside analog with broad spectrum activity against multiple positive and negative strand RNA viruses including beta-coronaviruses [27,28], RSV and paramyxoviruses [27], and both seasonal and highly pathogenic avian influenza viruses [29]. Inducing immediate chain termination of the IAV polymerase, 4'-FlU displayed an unusually wide therapeutic time window in the lethal pdm09 H1N1 mouse model, mediating complete survival of animals when treatment was initiated up to 60 hours after infection [29].

To explore the developmental potential of 4'-FlU for influenza therapy, we resistance profiled the compound in this study and assessed fitness and pathogenesis of resistant recombinants *in vitro* and *in vivo*, employing both the lethal mouse and ferret transmission model. We have identified three distinct clusters that induce moderate resistance to 4'-FlU. All three are predicted to affect the geometry of the central polymerase cavity that harbors the active site for phosphodiester bond formation. Resistance in all cases coincided with viral attenuation *in vitro* and *in vivo*. 4'-FlU treatment at the lowest efficacious (2 mg/kg) and/or moderately elevated (10 mg/kg) dose mediated complete survival after inoculation of animals with 10 $LD_{50}$ of the different resistant strains and suppressed viral transmission. These results define a high genetic barrier of 4'-FlU against CA09 escape from inhibition.

## Results

Dose-response assessment of 4'-FlU against recCA09 returned 50 and 90% inhibitory concentrations ($EC_{50}$ and $EC_{90}$, respectively) of 0.14 and 0.24 μM (S1 Fig), which was consistent with our previous observations [29].

### IAV adaption identifies distinct mutations mediating partial escape from 4'-FlU

Starting with an $EC_{90}$-equivalent compound concentration, we gradually adapted recCA09 to grow in the presence of 4'-FlU through dose-escalation serial passaging in six independent lineages (Fig 1A). At every passage, progeny virus titers were determined, viral RNA extracted, and fresh cells infected at a controlled multiplicity of infection (MOI) of 0.01 $TCID_{50}$ units/cell. Four additional adaptation lineages were carried out in the presence of vehicle (DMSO) volume equivalents. After 9–10 passages gradually exceeding 2.5 μM 4'-FlU, which was the sterilizing concentration of recCA09 at study start (S1 Fig), adaptation candidates replicated productively in the presence of 6 to 20 μM 4'-FlU (Fig 1B), whereas DMSO-experienced virus populations remained fully sensitive to 4'-FlU (Fig 1C). Whole genome sequencing of 4'-FlU-experienced virus populations at study end and selected intermediate passages identified in each lineage at least one, and up to three, point mutations in RdRP subunits that had become allele-dominant, but were absent in the vehicle-treated virus populations (Fig 1D; S1 Table; S1 Data). Two adaptation lineage pairs, #1/#5 and #4/#6, shared one common substitution, PB1

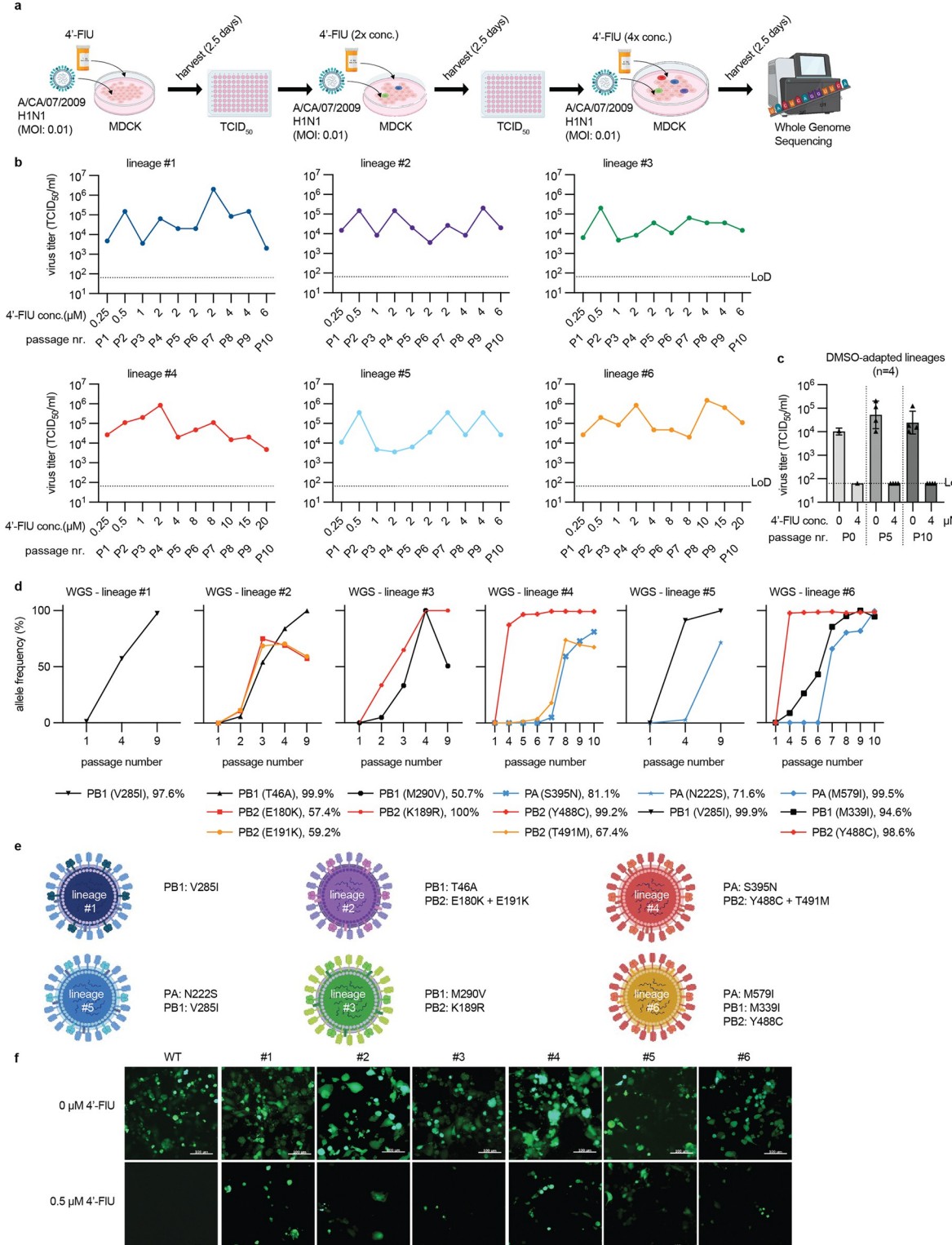

**Fig 1. Adaptation of CA09 to 4'-FlU. a)** Schematic of the adaptation strategy. Six independent lineages were generated in 10 consecutive passages. Clip-art was generated with BioRender. **b)** Virus titers after each passage (P1-P10). Corresponding 4'-FlU concentrations are shown for each passage and lineage. LoD, limit of detection. **c)** Virus titers of four vehicle (DMSO) volume equivalent adaptation lineages generated in parallel. Shown are titers after passages 0 (P0), 5 (P5), and 10 (P10). **d)** Whole genome sequencing (WGS) of adaptation lineages from (b). Shown are relative allele frequencies of mutations in RdRP subunits that reached >50% prevalence and were absent in

DMSO-experienced reference lineages. **e)** Color-coding schematic and substitutions in RdRP subunits in the six adaptation lineages after P10. Relative allele frequencies at adaptation end are shown in parenthesis. **f)** Fluorescence microphotographs of genetic parent recCA09-GFP (WT) and variants harboring the mutations shown in (e) after growth in the presence or absence of 4'-FlU. Scale bar, 100 μm.

V285I and PB2 488C, respectively (Fig 1E). Although not identical, lineages #2 and #3 harbored allele-dominant mutations in immediate vicinity of each other in PB2 (E191K and K189R, respectively). Other mutations were unique to the individual lineages. Individual mutations in three lineages (#4, #5, and #6) emerged with distinct dynamics, possibly indicating fitness compensatory effects of the substitutions appearing in later passages.

We rebuilt all mutations in polymerase subunits in a recCA09-maxGFP background, which expresses a maxGFP from the HA segment [29,30]. Infection of MDCK cells with the resulting recombinants in the presence or absence of 4'-FlU revealed different degrees of reduced susceptibility to the inhibitor compared to that of the genetic parent virus (Fig 1F and S2 Fig).

### Three pathways to achieve viral resistance to 4'-FlU

To quantify robustness of viral resistance in an otherwise unchanged viral background, we rebuilt in parallel a set of non-reporter recCA09 variants. Dose-response assays against 4'-FlU demonstrated a 2 to 25-fold increase in $EC_{99}$ concentrations of 4'-FlU against the different virus lineages, confirming moderate resistance to 4'-FlU (Fig 2A and S2 Table). A favipiravir-resistant recCA09 carrying the PB1 K229R and PA P653L point mutations previously described [26] showed unchanged sensitivity to 4'-FlU. The inverse experiment, testing the recCA09 panel against nucleosides analogs currently in clinical use for respiratory virus indications, favipiravir and molnupiravir [31,32], revealed slightly decreased (favipiravir) or increased (molnupiravir) viral susceptibility, but only lineage #4 showed cross-resistance with favipiravir that reached an $EC_{99}$ change >10-fold (Fig 2B and 2C; S3 and S4 Tables).

Whereas maximal growth rates of resistant viruses were unchanged to that of their genetic parent in multi-step growth curves on MDCK cells, maximal progeny titers reached were reduced by up to 1.5 orders of magnitude (Fig 2D and S5 Table). To directly compare relative fitness of mutants and their genetic parent, we co-infected cells with a 10:1-mixture of mutant to parent, followed by five serial passages with whole genome sequencing of the viral populations after each passage (Fig 2E). Relative allele frequencies of mutations in lineages #1, #2, and #3 plummeted rapidly, indicating a fitness penalty compared to the parent virus (Fig 2F and S2 Data). Lineage #4 co-replicated with the genetic parent without substantial change in relative allele frequencies, and one substitution each of lineages #5 and #6, V285I in PB1 and Y488C in PB2, respectively, disappeared rapidly, whereas others co-existed with the parent allele.

To better differentiate the contribution of discrete mutations to the resistance phenotype, we rebuilt all substitutions individually in recCA09 reporter virus expressing a nano luciferase from the HA segment (recCA09-nanoLuc), followed by luciferase-based dose-response assays (Fig 2G; Table 1; S3 Fig). Three different pathways to viral escape emerged: i) individual mutations caused resistance (i.e. PB1 V285I in lin #1 and #5), sometimes in combination with an enhancer (lin #5); ii) resistance was based on additive effects (i.e. lin #3); or iii) resistance arose from synergy (i.e. lin #2, #4, and #6). In one case (lin #4), the individual role of one mutation (PB2 T491M) could not be assessed, since this substitution was viable only in the context of the PB2 Y488C exchange (Table 1).

None of the combinations of mutations required for moderate resistance to 4'-FlU existed in complete IAV genome sequences available in the NCBI Virus database (S4 Fig). Although some mutations were present individually, all remained below 1% relative frequency except

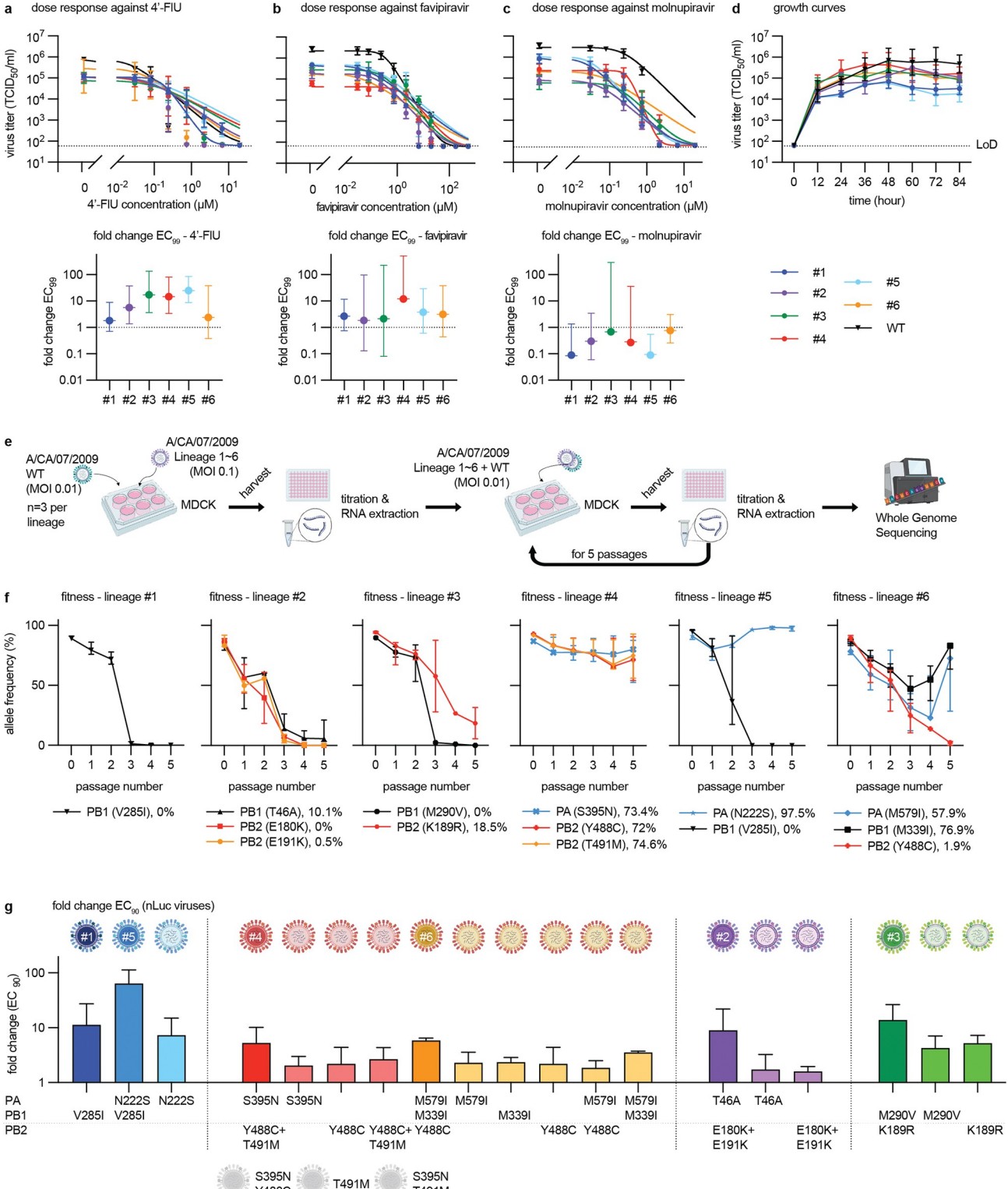

**Fig 2. Validation of moderate resistance to 4'-FlU through engineered recCA09. a-c)** Virus titer reduction-based dose response assays of six recCA09 harboring candidate mutations from (Fig 1E) against 4'-FlU (a), favipiravir (T-705) (b), and the active metabolite of molnupiravir (EIDD-1931) (c). Upper panels: dose response assay results. Symbols represent geometric means ± geometric SD, lines show 4-parameter variable slope regression models. Bottom panel: fold change EC99 relative to parental recCA09. Symbols show means, whiskers denote 95% confidence intervals (CI); n = 3. **d)** Growth curves of the rebuilt recCA09 adaptation lineages and parental recCA09. Symbols show geometric means ± geometric SD, lines connect means. **e)**

Schematic of *ex vivo* assessment of relative viral fitness through coinfection with parental recCA09 in a 10:1 ratio. Clip-art was generated with BioRender. **f)** Whole genome sequencing of co-infection populations from (e). Relative allele frequencies of resistance mutations in RdRP subunits are shown after each of five consecutive passages. Symbols represent medians ± 95% confidence intervals (CI), lines connect medians; n = 3. Next generation sequencing over time. Lineages 1 to 6 with co-infected WT are displayed. **h)** Re-validation of resistance mutations singly and in combination in recCA09-nanoLuc reporter virus background. Fold-change of $EC_{90}$ values relative to parental recCA09-nanoLuc are shown. Regression models were developed based on luciferase dose-response assays. Columns show mean $EC_{90}$ values + SD; n = 3; greyed virion schematics denote mutation combinations that were non-recoverable.

PB1 M339I and PB2 E191K in complete and partial sequences (S6 Table), confirming our previous observation that circulating human and avian IAVs are efficiently inhibited by the compound [29].

## Spatial localization implicates resistance mutations with altered RdRP geometry

We localized all resistance mutations in structural models of the influenza virus polymerase complex, built on the coordinates released for an avian H5N1 IAV [33], the 1918 H1N1 IAV [34], an influenza virus C (ICV) [35], and a bat IAV [36] (Figs 3A and S5A–S5C). The spatial positions of the mutations were virtually identical in the H5N1 (Fig 3A), 1918 H1N1 (S5A Fig), and ICV structures, which are all considered to represent a pre-initiation conformation of the polymerase. However, mechanistic assessment was based on the H5N1 structure, which encompasses the intact RdRP complex of an IAV, whereas the 1918 H1N1 IAV model lacks parts of the PA and PB2 subunits and the bat IAV represents an alternate cap-binding

**Table 1. Polymerase mutations that emerged in adaptation to 4'-FlU, rebuilt as emerged (lineages #1–6) and individually or sub-combinations (sub-lineages #II-VI) in recCA09.**

| adaptation lineage | PA | PB1 | PB2 | $EC_{90}$ [μM] | predicted effect |
|---|---|---|---|---|---|
| #1 | | V285I | | 16.8 μM | causal |
| #5 | N222S | V285I | | 49.4 μM | |
| V-1 | N222S | | | 3.9 μM | enhancer |
| #2 | | T46A | E191K + E180K | 6.5 μM | |
| II-1 | | T46A | | 0.4 μM | synergistic |
| II-2 | | | E191K + E180K | 1.3 μM | synergistic |
| #3 | | M290V | K189R | 13.2 μM | |
| III-1 | | M290V | | 3.5 μM | additive |
| III-2 | | | K189R | 5.2 μM | additive |
| #4 | S395N | | Y488C + T491M | 7.2 μM | |
| IV-1 | S395N | | | 0.9 μM | synergistic |
| IV-2 | | | Y488C + T491M | 3.4 μM | synergistic |
| IV-3[A] | S395N | | Y488C | revertant to WT | |
| IV-4[A] | | | T491M | revertant to WT | |
| IV-5[A] | S395N | | T491M | no recovery | |
| #6 | M579I | M339I | Y488C | 5.8 μM | |
| VI-1 | | | Y488C | 0.7 μM | synergistic |
| VI-2 | M579I | M339I | | 3.7 μM | synergistic |
| VI-3 | M579I | | | 1.0 μM | synergistic |
| VI-4 | | M339I | | 1.3 μM | synergistic |
| VI-5 | M579I | | Y488C | 1.1 μM | synergistic |

[A] not recoverable

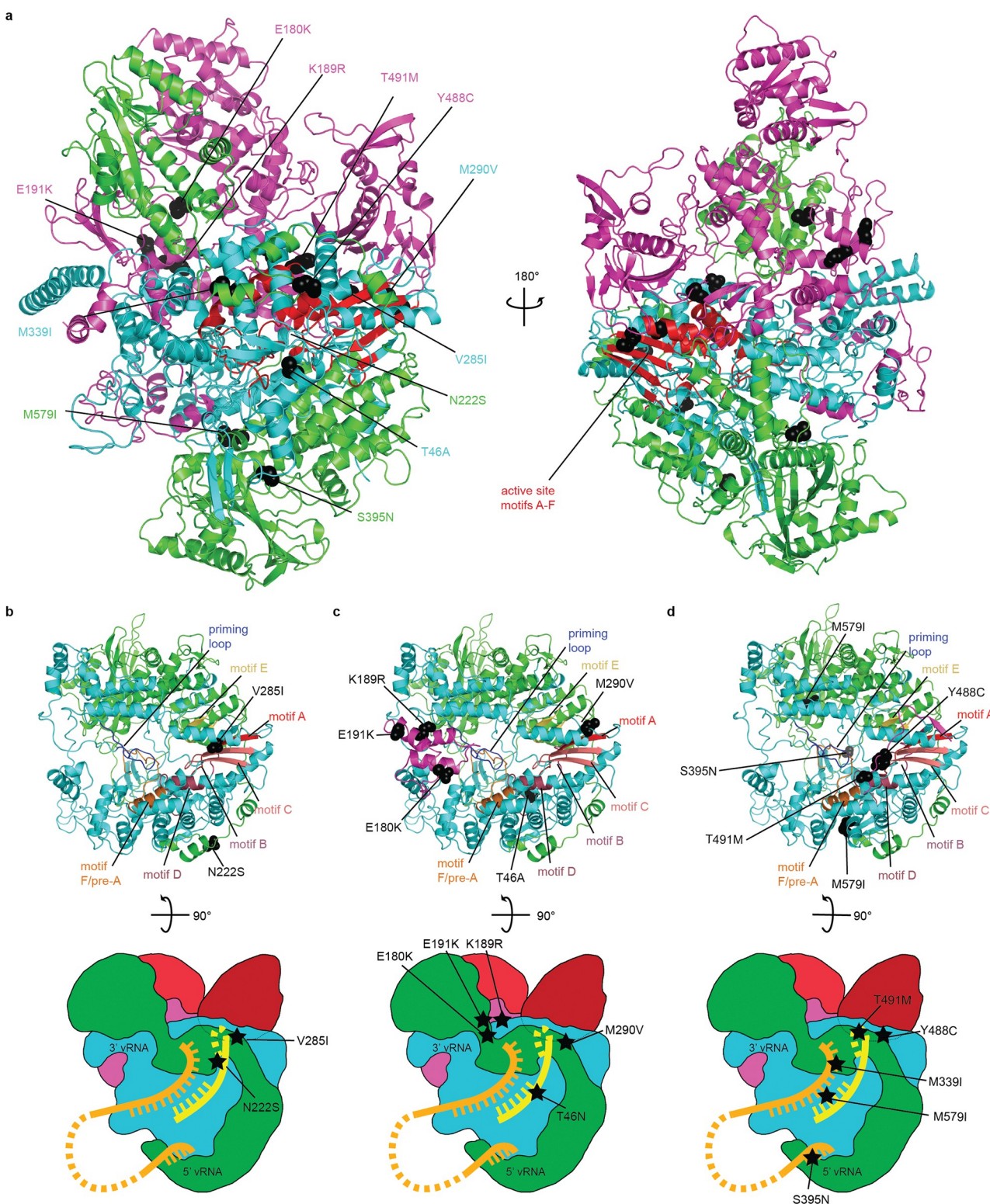

**Fig 3. Structural mechanism of CA09 resistance to 4'-FlU. a)** Overview of the locations of resistance mutations identified in viral adaptation to growth in the presence of 4'-FlU. Mutation denotations are color-coded by polymerase subunit (PA = green, PB1 = cyan, PB2 = magenta). The active site of the RdRP complex for phosphodiester bond formation is shown in red. **b-d)** Ribbon representations (top) and polymerase schematics (bottom) highlighting the mutations associated with the distinct paths to resistance identified for lineages #1/5 (b), #2/3 (c), and #4/6 (d). Motif A-F [54] of the IAV RdRP are shown; orange, template RNA strand; yellow, nascent RNA strand. Homology models of the CA09 polymerase were created using SWISS-MODEL based on PDB ID 6QPF; images were created using Pymol.

conformation. Most mutations were posited in the central RdRP cavity, either in immediate proximity to the catalytic site for phosphodiester bond formation (lin. #1 and 5; Fig 3B and lin. #4 and 6; Fig 3C) or at positions in short-range proximity to the GDN active site (lin. #2 and 3; Fig 3D). Noticeable exceptions were the PB2 E180K/E191K and PB2 K189R charge-reversal substitutions in lin. #2 and lin. #3, respectively, which may affect structural mobility of PB2 and may alter cap RNA loading into the central cavity.

Overall, altered geometry of the polymerase active site, achieved through combinations of mutations in distinct sets, emerged as the common theme across all resistance lineages (S7 Table). These changes may either increase selectivity of the polymerase complex for substrate triphosphates or, more likely based on *in vitro* RdRP assay results of immediate chain termination [29], increase spatial flexibility at the active site to better accommodate secondary structure changes introduced by incorporation of the 4'-FlU analog into the nascent RNA strand [27,29]. Structural limits for flexibility appear to be tight, however, since different artificially engineered combinations of independently emerged signature resistance mutations did not result in bioactive polymerase complexes that could support productive virus replication (S8 Table).

These findings pinpoint residues defining the geometry of the central RdRP cavity as instrumental for moderately reduced polymerase susceptibility to inhibition by 4'-FlU, and suggest that the genetic barrier preventing emergence of more robust resistance may be major.

## All 4'-FlU-experienced resistant variants are attenuated *in vivo*

In the Balb/cJ mouse model, infection of animals with equal infectious particle amounts of the resistant variants or parent recCA09 followed by daily monitoring of clinical signs provided no indication of enhanced disease (S6 Fig). To compare pathogenicity with parent recCA09, we determined $LD_{50}$ inoculum amounts (Fig 4A). Relative to recCA09, all resistant strains were attenuated, ranging from 25-fold (lin. #4) to 1,000-fold (lin. #5) higher $LD_{50}$ values (Fig 4B–4D). Despite the fitness disadvantage *in vitro* and attenuation *in vivo*, genetic stability of the resistant strains was high in mice in the absence of compound (Fig 4E). Relative allele frequency of all mutations but PB1 M290V was near 100% in virus populations isolated from lung tissue 5 to 10 days after infection (S3 Data), when individual animals reached endpoints.

To explore the effect of resistance on *in vivo* drug efficacy, animals were infected with $10 \times LD_{50}$ equivalents of each variant, followed 24 hours after infection by five doses of 4'-FlU at 2 (lowest efficacious standard dose) or 10 (elevated dose) mg/kg bodyweight, administered orally q.d. (Fig 4F). Animals in survival groups were monitored for 14 days, lung virus loads were determined in subgroups 4.5 days after infection. Standard dose resulted in complete survival of animals infected with parent recCA09 or all resistant variants except lin. #5, which showed 50% survival (Figs 4G, 4H and S7). All vehicle-treated animals succumbed by day 4 after infection. Elevated dose of 4'-FlU resulted in complete survival also of animals infected with lin. #5. Lung virus loads mirrored survival data, showing significant reductions in all cases after both standard and elevated dose of 4'-FlU (Figs 4H and S8). However, effect size was greatest in the case of parent recCA09 and lin. #2 (undetectable after elevated dose) and least pronounced in animals infected with lin. #5.

These data demonstrate that resistant variants are attenuated, albeit still pathogenic in the mouse model. However, moderate resistance to 4'-FlU can be overcome pharmacologically *in vivo*.

## 4'-FlU prevents transmission of resistant variants in the ferret model

Two resistant variants, lin. #4 and lin. #5, were selected for direct-contact transmission studies in the ferret model based on highest pathogenicity and greatest resistance to 4'-FlU,

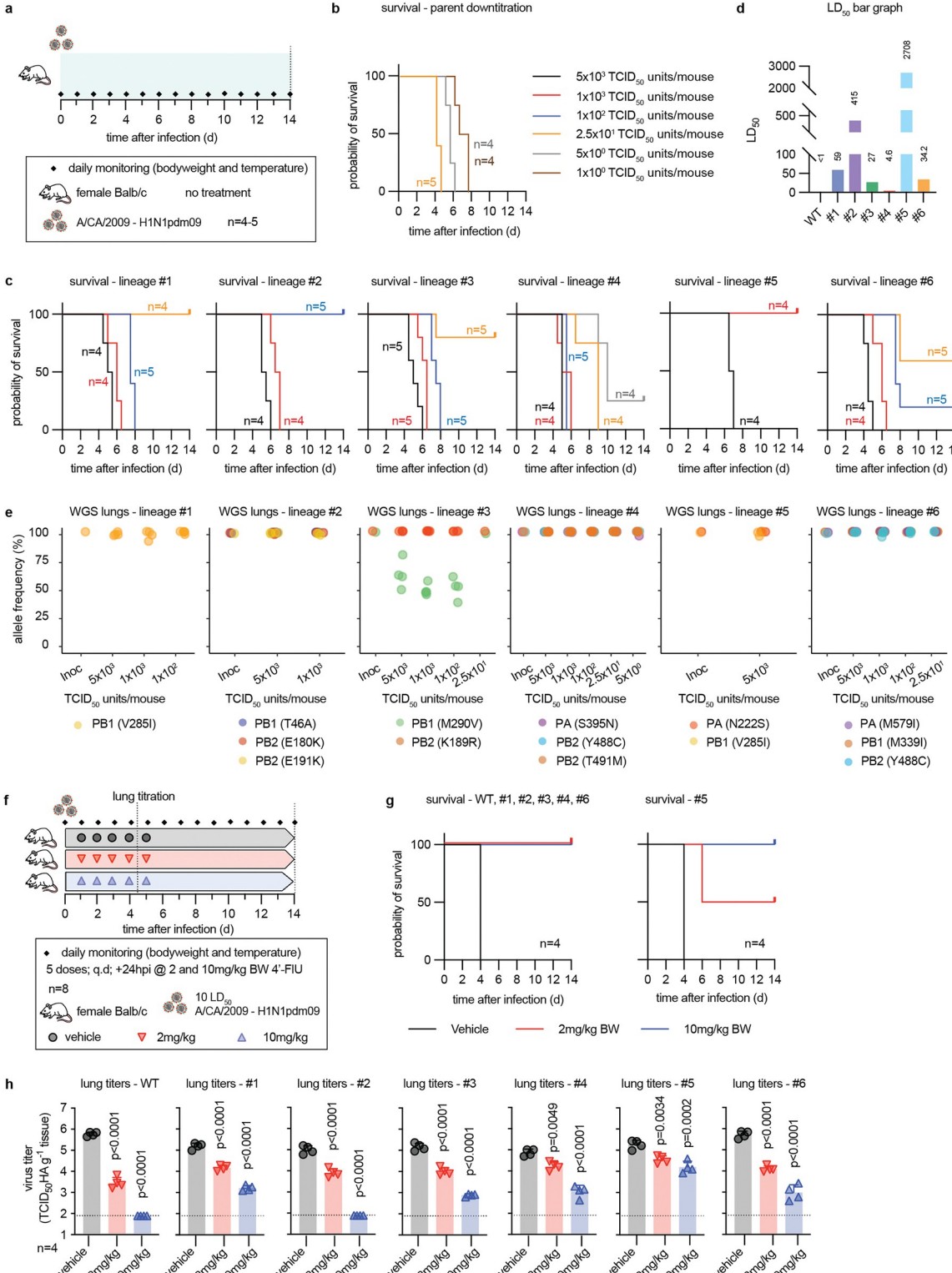

**Fig 4. 4'-FlU-resistant recCA09 are attenuated in mice. a)** Schematic of the pathogenicity study. Mice were monitored for 14 days or until predefined endpoints were reached. **b-d)** Assessment of LD$_{50}$ nasal inoculum. Survival curves after infection with varying inoculum amounts of parent recCA09 (WT) (b) and resistant recCA09 (c) were used to calculate LD$_{50}$ values (d) through Mantel-Cox analysis; n numbers as specified. **e)** Whole genome sequencing of virus populations recovered from mouse lungs at endpoint. Relative allele frequencies of resistance mutations at the time of infection (Inoc) and endpoint for each of the different inoculum groups are

shown. Symbols represent virus populations recovered from individual animals. **f)** Schematic of *in vivo* efficacy study. Mice were infected with 10 $LD_{50}$ units of parent or resistant recCA09; treatment with 4'-FlU at standard (2 mg/kg) and elevated (10 mg/kg) oral dose was initiated 24 hours after infection. **g)** Survival curves. Shown are results for parental recCA09 (WT) and lineages #1/2/3/4/6 (left) and parental recCA09 and lineage #5 (right); n = 4. **h)** Lung virus load 4.5 dpi of animals from (f). Symbols represent individual animals, columns show geometric means + geometric SD. Statistical analysis with 1-way ANOVA and Dunnett's multiple comparison post-hoc test; p values are stated; n = 4.

respectively. Treatment of source ferrets with three oral doses of 4'-FlU at 2 mg/kg bodyweight each was initiated 12 hours after infection and 36-hour\co-housing started 2.5 dpi, after administration of the last compound dose (Fig 5A). Source animals were euthanized at the end of the co-housing period and contacts monitored until study day 8.

Treatment of source animals with 4'-FlU alleviated clinical signs in all study arms (S9 Fig). recCA09 was first detectable in nasal lavages of all sentinels of vehicle-treated ferrets at the end of the contact period (Fig 5B), confirming efficient IAV transmission in the ferret model [29]. Consistent with our previous observations [29], 4'-FlU was highly efficacious of reducing shed virus load in the treated source animals, resulting in a complete block of transmission. Peak shed virus load of recCA09 lin. #4 was reduced in vehicle-treated animals by approximately one order of magnitude compared to parental recCA09 (Fig 5C), whereas recCA09 lin. #5 reached similar peak titers (Fig 5D). However, recCA09 lin. #4 efficiently spread from vehicle-treated source to contact animals, while recCA09 lin. #5 was transmission-incompetent. Oral treatment of recCA09 lin. #4-infected animals with standard dose 4'-FlU completely blocked transmission, suggesting that moderate resistance does not compromise pharmacological control of IAV.

Nasal turbinate (Fig 5E), trachea (Fig 5F), and lung (Fig 5G) titers determined at study end reflected the efficient block of parent and mutant recCA09 transmission by 4'-FlU. Independent of inoculum variant, virus was undetectable in all tissues harvested from sentinels of 4'-FlU-treated source animals. Treatment furthermore suppressed viral invasion of the lower respiratory tract in animals infected with recCA09 (Fig 5F and 5G). In contrast, trachea load of vehicle-treated animals infected with recCA09 lin. #4 or #5 was reduced by approximately 2.5 orders of magnitude compared to parental recCA09 and viral pneumonia was alleviated in the absence of treatment, confirming attenuation of these variants.

Whole genome sequencing of the different virus populations extracted from animal lungs at study endpoints revealed genetic stability of recCA09 lin. #4 through the course of the study in both source and, after successful transmission from vehicle-treated animals, contact ferrets (Fig 5H; S4 Data). However, both mutations present in the recCA09 lin. #5 inoculum, PA N222S and PB1 V285I, were rapidly counter-selected against in ferrets, resulting in relative allele frequencies of approximately 5 and 25%, respectively, in the vehicle-treated source animals at 4 dpi. Virus load in all other recCA09 lin. #5 groups was too low to support a reliable whole genome analysis.

These data demonstrate that moderate resistance to 4'-FlU is achievable, but in all cases coincided with viral attenuation represented by alleviated clinical signs, reduced tissue virus burden, and impaired invasion of the lower respiratory tract. Even transmission of variants with the most robust resistance phenotype or greatest residual pathogenicity is compromised an/or readily controlled by standard dose 4'-FlU.

## Discussion

Pre-existing or rapidly emerging widespread resistance to antiviral countermeasures, both small-molecule drugs and biologics, has undermined antiretroviral monotherapy [37], invalidated the use of first-generation therapeutic antibodies against SARS-CoV-2 [38], negatively

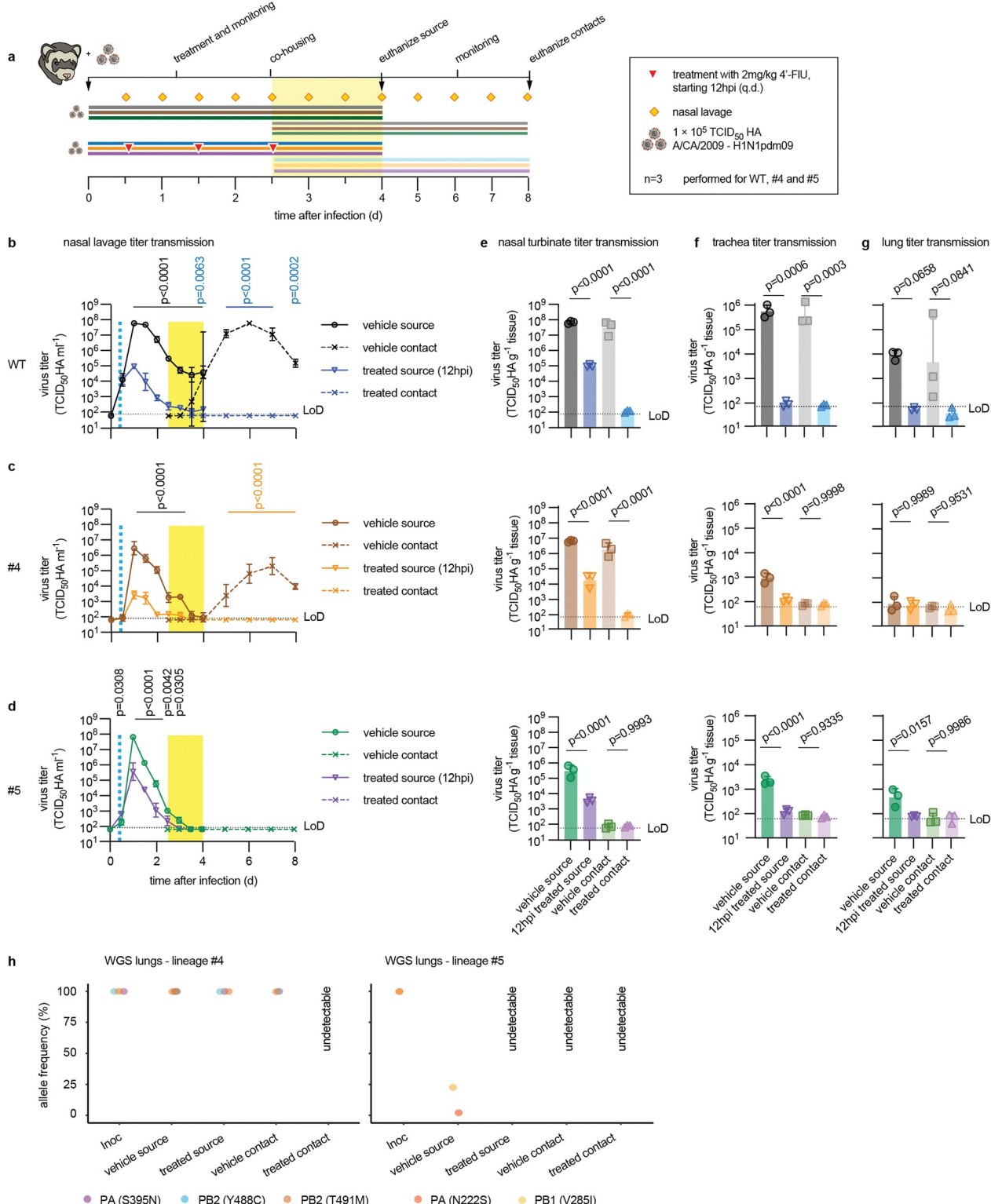

**Fig 5. 4'-FlU-resistant CA09 are transmission-impaired in ferrets. a)** Schematic of the direct contact transmission study, assessing parent recCA09 (WT) and rebuilt, recombinant CA09 representing lineages #1/5 and #4/6. **b-d)** Nasal lavage titers of source (solid lines) and contact (dashed lines) animals infected with parental recCA09 (b), recCA09 lineage #4 (c), or recCA09 lineage #5 (d). Vertical dashed blue line indicates start of treatment of source animals, yellow bar denotes time of co-housing. Symbols represent geometric means ± geometric SD, lines connect means. 2-way-ANOVA with Sidak's multiple comparison post-hoc test; p values are stated. **e-g)** Virus load in nasal turbinates (e), trachea (f) and

lung (g) of animals from (a), determined 4 and 8 dpi for source and contact animals, respectively. Symbols represent individual animals, columns show geometric means ± geometric SD; 1-way ANOVA with Tukey's multiple comparison post-hoc test; p values are stated. **h)** Whole genome sequencing of virus inoculum (Inoc) and virus populations extracted from animals from (a) at study endpoint when possible. Relative allele frequency cut-off 3%; symbols represent individual animals.

impacted COVID-19 vaccine efficacy [39], destroyed efficacy of one of three FDA-approved influenza drug classes [17,18], and compromises the benefit of a second influenza drug class [40]. Early profiling of the viral escape landscape is therefore essential in drug development. Although there is little precedent for antivirals inducing resistant variants with increased pathogenicity, the prospect of triggering enhanced disease could be catastrophic in the clinic and must be examined prior to formal preclinical development.

The broad-spectrum nucleoside analog inhibitor 4'-FlU is a clinical candidate for different RNA virus indications [27,29]. If ultimately FDA approved for influenza therapy, the compound will represent a first-in-class, expanding the anti-influenza virus arsenal to four distinct drug types. Compared to allosteric small-molecule drugs, nucleoside analogs typically show a higher genetic barrier against viral escape [41]. Influenza virus escape from the viral mutagen favipiravir can be achieved *ex vivo* through a combination of a causal mutation mediating escape and a compensatory mutation that partially restores viral fitness [26]. These favipiravir-resistant viruses were transmission-competent in the ferret model [42], but their susceptibility to favipiravir treatment was not tested *in vivo* and no resistance to favipiravir emerged in clinical trials [43,44]. Our IAV adaptation study to 4'-FlU supports five major conclusions:

i. CA09 escape from 4'-FlU inhibition is possible, but the genetic barrier to escape is high and robust resistance, defined as an increase in inhibitory concentration by >100-fold, could not be achieved. This result underscores that it is challenging to increase selectivity of the viral polymerase complex for endogenous nucleotides over analog inhibitors without compromising overall RdRP bioactivity. In our previous *in vivo* efficacy studies with 4'-FlU, we did not observe any compound-induced changes in viral polymerase components when 4'-FlU-experienced virus populations reisolated from infected and treated animals were subjected to whole genome sequencing [27,29]. Although one cannot definitively address the question of whether our *ex vivo* adaptation study was exhaustive, the redundancies of signature resistance mutations across independent adaptation lineages and the paucity of additional mutations arising from viral escape *in vivo* strongly support that major hot-spots of IAV resistance to 4'-FlU have been appreciated in our study.

ii. Remarkably, three distinct avenues to partial resistance to 4'-FlU were available to CA09: a single causal substitution, optionally combined with an enhancer; additive action of mutations of minor phenotypic effect in isolation; and synergy between substitutions that in isolation do not alter sensitivity to 4'-FlU. The viral fitness landscape for some of the synergistic mutations was extremely narrow, since rebuilding them in isolation (i.e. PB2 T491M) or in subsets (i.e. PB2 Y488C with PA S395N or PB2 T491M with PA S395N) did not result in viable polymerase complexes. The dynamics of gradual acquisition of these mutations during viral adaptation and their disappearance during *ex vivo* co-culture with standard recCA09 are consistent with strong interdependence of specific combinations for RdRP bioactivity. Precedent for synergistic effects of individual mutations leading to reduced viral drug susceptibility was established by drug resistant HIV [45], SARS-CoV-2 escape from protease inhibitors [46], and reduced IAV resistance to neuraminidase [47,48] and experimental entry [49] inhibitors. Exploitation of three distinct mechanistic pathways to reduce viral susceptibility to inhibition is unique to 4'-FlU, however, and suggests that a dominant hot spot of viral escape that results in robust resistance does not exist. Multiple

avenues for viral escape furthermore distinguish 4'-FlU from favipiravir, for which only a single combination carrying resistance was previously identified [26]. Our work expanded the favipiravir resistance panel by a novel combination of substitutions that provided moderate escape from favipiravir. However, introducing the previously identified favipiravir escape mutation [26] into the recCA09 background did not result in resistance to 4'-FlU. Resistance to 4'-FlU did furthermore not coincide with reduced sensitivity to broad-spectrum molnupiravir, establishing a promising foundation for combination therapies with nucleoside analog inhibitors.

iii. Although mechanistically distinct, escape mutations of all lineages located predominantly to the central cavity of the RdRP complex, either in direct proximity to residues contacting the RNA template and nascent strands (i.e. lineages #1, 4, 5, 6) or at domains predicted to affect the spatial orientation of these residues through long-range structural effects. In all cases, these substitutions are predicted to alter the symmetry of the central polymerase cavity, most likely increasing tolerance of the RdRP complex to accommodate secondary structure changes introduced by addition of 4'-FlU to the nascent RNA strand. We have demonstrated previously that 4'-FlU inhibits IAV RdRP by immediate chain termination, but acts as a delayed or sequence context-dependent delayed chain terminator on RSV and SARS-CoV-2 polymerases, respectively [27,29]. These changes were reflected by corresponding potency differences of 4'-FlU against these viral targets in cell culture. Coincidentally, attempts to resistance profile RSV against 4'-FlU have been unsuccessful to date. We hypothesize that IAV RdRP may have access to a fitness landscape of moderately increased structural flexibility in the central RdRP cavity while maintaining bioactivity, whereas tolerance of RSV and SARS-CoV-2 to accommodate secondary structure changes cannot be further expanded without loss of polymerase function. This notion implies that a hard ceiling of reducing susceptibility to 4'-FlU exists also for IAV RdRP that cannot be broken without eliminating all bioactivity. If correct, this hypothesis implies that an equivalent path to moderate resistance as that mapped for CA09 may be available for unrelated viruses susceptible to 4'-FlU, provided the compound inhibits the polymerases of these viruses through direct chain termination.

iv. Resistant variants are attenuated *in vivo*, albeit not apathogenic, and remaining clinical signs can be mitigated with 4'-FlU therapy. A primary objective of resistance profiling a novel developmental candidate is to establish whether viral escape may coincide with increased pathogenicity. Although rare in response to small-molecule drugs, any risk of triggering enhanced disease would halt further preclinical and clinical development of a compound. Our study returned a clean escape profile for 4'-FlU without any evidence of exacerbated disease by resistant variants in a rodent and non-rodent model. Since most resistance mutations also reduced viral fitness compared to the genetic parent virus in cell culture, reduced replication efficiency is the most likely reason for viral attenuation *in vivo*. To our surprise, only the least pathogenic resistant variant, based on $LD_{50}$, could not be fully controlled *in vivo* by standard dose 4'-FlU. The data provide strong evidence, however, that viral escape will not undermine sustained therapeutic benefit of 4'-FlU, since a moderate increase in dose levels fully suppressed clinical signs caused by this variant and resulted in complete survival of infected animals. This feature distinguishes 4'-FlU from non-competitive IAV inhibitors such as baloxavir marboxil and oseltamivir, that do not show a strong direct correlation between resistance and virus fitness penalty.

v. 4'-FlU resistant variants are transmission-impaired, and spread of those with residual transmission competence can be suppressed with 4'-FlU treatment. Two constellations pose a

particular risk to trigger the development of wide-spread pre-existing resistance to an anti-viral in circulating viruses: an escape mutation that provides a transmission advantage; or a combination of saturating selective pressure on circulating viruses with unaltered transmissibility of resistant variants. The former drove, for instance, the evolution of SARS-CoV-2 variants of concern with increasingly greater transmissibility [39] and the latter is best exemplified by large scale use of the adamantes in the poultry industry, which destroyed clinical benefit of the M2 channel blockers [15,16]. Neither of these constellations appears to be met in the case of 4'-FlU, since signature resistance mutations of adaptation lineage pair #1/5 interrupted transmission competence and spread of moderately resistant variants of the lineage #4/6 pair could be readily suppressed with 4'-FlU treatment. These results suggest that clinical use of 4'-FlU would be unlikely to promote evolution of circulating IAV strains, thus setting the stage for long-term therapeutic benefit.

Although results in cell culture and rodent and non-rodent *in vivo* models of IAV infection support that the 4'-FlU resistance profile determined in this study is not limited to a specific experimental system, it is currently unknown whether *ex vivo* and animal model-based results of viral attenuation equally apply to the human host. Other limitations of the study include that individual resistance mutations could be IAV strain and/or subtype specific, that the specific structural state of the RdRP complex targeted by the inhibitor is unknown, and that human dose levels, and therefore by extension drug tissue exposure in the clinic, are currently undetermined. These precautions notwithstanding, this study defines mechanisms reducing susceptibility of the IAV polymerase complex to a chain-terminating nucleoside analog inhibitor and demonstrate in rodent and non-rodent animal models that CA09 influenza virus escape from the developmental drug 4'-FlU is mandatorily linked to viral attenuation and reduced transmission competence. These data suggest that 4'-FlU resistant virus populations are unlikely to reach clinical significance or persist in circulation in the field, supporting the long-term therapeutic potential of the compound.

## Material and methods

### Ethics statement

All animal work was performed in compliance with the *Guide for the Care and Use of Laboratory Animals* of the National Institutes of Health and the Animal Welfare Act Code of Federal Regulations. Experiments involving mice and ferrets were approved by the Georgia State University Institutional Animal Care and Use Committee (IACUC) under protocols A20012 and A21020, respectively. All experiments using infectious material were approved by the Georgia State University Institutional Biosafety Committee (IBC) and performed in BSL-2/ABSL-2+ containment facilities.

### Study design

Cells, mice, and ferrets were used as *in vitro* and *in vivo* models to examine the resistance profile of 4'-FlU against influenza viruses. *In silico* modelling was added for mechanistic characterization. Viruses were administered through intranasal inoculation and virus load monitored periodically in nasal lavages (ferrets only), and in respiratory tissues of ferrets and mice extracted 4 days (mice and ferrets) and 8 days (ferret contacts) after infection. Virus titers were determined through 50% tissue culture infectious dose ($TCID_{50}$) titration.

### Cells and viruses

HEK293T human kidney cells (293T; ATCC CRL-3216) and Madin-Darby canine kidney cells (MDCK; ATCC CCL-34 were maintained at 37°C and 5% $CO_2$ in Dulbecco's Modified Eagle's

medium (DMEM) supplemented with 7.5% fetal bovine serum (FBS). Mammalian cell transfections were performed using GeneJuice transfection reagent (Invitrogen). recCA09 and recCA09-based reporter viruses such as recCA09-maxGFP were recovered from cloned cDNA and amplified as previously described [29,30]. Sequences of recombinant virus stocks were identical to those reported for A/CA/07/2009 (H1N1) under NCBI:txid641809 with the exception of 2 point mutations in NP (D101G and L122Q) that have also been seen in unrelated pdm09 IAV (H1N1) isolates [50,51]. The NP D101G substitution was furthermore associated with mouse adaptation of closely related A/CA/04/2009 (H1N1) [51], providing a likely explanation for the high mouse pathogenicity of the recCA09 used in this study.

## TCID$_{50}$ titration

For virus titrations, MDCK cells were seeded in 96-well plates ($1.5 \times 10^4$ cells/well) in 8 replicates/dilution tested. Virus samples were serially diluted 10-fold, and diluents transferred to each well. After incubation at 37°C for 72 hours, viral presence in culture supernatants was determined by hemagglutination assay using 0.5% chicken erythrocytes or direct assessment through fluorescence microscopy in the case of GFP-expressing recCA09 reporter viruses. Raw data were analyzed by the Reed and Muench method.

## Virus adaptation

MDCK cells were infected with IAV recCA09-maxGFP at an MOI of 0.01 TCID$_{50}$ units/ml and incubated with 0.25 μM of 4'-FlU or DMSO. Green fluorescence was monitored as a marker for virus replication and virus populations were harvested based on fluorescence saturation. Virus populations were titered after each passage. Fresh MDCK cells were infected with the harvested virus populations at an MOI of 0.01 TCID$_{50}$ units/cell in the presence of 4'-FlU at specified concentrations or vehicle (DMSO) volume equivalents. Each passage was MOI controlled and the drug concentration kept fixed or doubled for the subsequent passage depending on visual assessment of fluorescence intensity. Viral RNA was isolated from different passages of all lineages for whole genome sequence analysis.

## Sequencing

For the determination of viral RNA from stocks and supernatants/organ homogenates, Quick-RNA Viral Kit (Zymo Research) was used and cDNA copies were generated using random hexamer primers and SuperScript III reverse transcriptase (Invitrogen). To identify mutated genomic regions, RNA extracts of virus populations were subjected to Sanger and/or whole genome sequencing as specified.

## qPCR for CA09 matrix and hemagglutinin

RT-qPCR was used to measure IAV RNA abundance. Primers and probes were designed to hybridize to the matrix or hemagglutinin sequences. AgPath-ID One-Step RT-PCR reagents (Thermo AM1005) kit was used according to manufacturer's instructions. Amplifications were carried out on a QuantStudio5 (Thermo # A34322) instrument.

## Sequencing protocol selection

All viral samples derived from *in vitro* experiments were sequenced by shotgun metagenomics. All samples derived from *in vivo* studies were sequenced using hybridization capture (only exception sample "ferret WT-C1-1: wildtype untreated contact replicate 1", which was

sequenced by amplification of the genomic RNA segments with primers specific for the homologous termini of each genomic segment).

## Metagenomic cDNA synthesis

Metagenomic sequencing libraries were prepared by DNAse-treatment of RNA extracts, followed by double-stranded cDNA synthesis, bead-linked tagmentation and adapter-ligation of double-stranded cDNA, cycle-limited amplification of tagmented cDNA with primer pairs that contain unique 10nt indexes. The indexed libraries were normalized and pooled for $1 \times 101$ or $2 \times 151$ cycle sequencing. Before cDNA synthesis, RNA extracts were treated with Turbo DNAse (Thermo #AM1907). After DNA digest, first-strand cDNA synthesis was performed using SuperScript IV (Thermo #18090010) and random hexamers (Thermo #N8080127).

Subsequent second-strand synthesis was performed using the Sequenase 2.0 kit (Thermo #70775Z1000UN) and amplicons cleaned with $1.8 \times$ AMPure XP magnetic beads (Beckman Coulter #A63882). Purified cDNA samples were stored at -20°C until library preparation.

## Metagenomic DNA library preparation

Metagenomic libraries were prepared from the double-stranded cDNA using the Illumina DNA Prep with Enrichment Kit (Illumina #20025524) and IDT for Illumina DNA/RNA UD Indexes Set D (Illumina #20042667). Concentrations of $1.8 \times$ AMPure XP magnetic bead purified libraries were measured using Qubit 1× dsDNA High Sensitivity Assay Kit (Thermo #Q33231) and a Qubit Flex Fluorometer (Thermo #Q33327). Library concentrations were normalized and libraries were pooled. The average library size for the pool was determined using an Agilent DNA D1000 Tape Station kit (Agilent #5067).

## Illumina respiratory virus hybridization capture

Hybridization capture was performed with an Illumina RNA Prep with Enrichment (L) Tagmentation kit (Illumina #20040537), IDT for Illumina DNA/RNA UD Indexes (Illumina #20027213), and Respiratory Virus Oligo Panel v2 (Illumina #20044311). Tagmented cDNAs were amplified to add indexes and adapters, and the resulting libraries normalized using Qubit for 1- or 3-plex enrichment by hybridization to sequence-specific biotinylated probes. The captured sequences were washed, eluted, and amplified to generate copies of the enriched libraries, which were further amplified and cleaned with AMPure XP Beads, normalized, and pooled for $1 \times 101$ or $2 \times 151$ cycle sequencing on an Illumina NovaSeq or NextSeq2000 instrument.

## IAV whole genome sequencing by amplification

Three primers that hybridize to the 12 nucleotides at the 3' end and 13 nucleotides at the 5' end of each IAV genomic RNA segment were used to amplify the viral genome: Uni12/Inf-1 (5′-GGGGGGAGCAAAAGCAGG-3′), Uni12/Inf-3 (5′-GGGGGGAGCGAAAGCAGG-3'), and Uni13/Inf-1 (5′-CGGGTTATTAGTAGAAACAAGG-3′) [52]. One-step RT-PCR was performed using the Superscript III One-Step RT-PCR System with HiFi Platinum Taq DNA polymerase (Thermo #12574035). After amplification, samples were cleaned with $0.8 \times$ AMPure XP Beads (Beckman Coulter #A63882) and products fractionated on a 1.2% Lonza FlashGel to visualize sizes. Of the cleaned PCR product, 25–100 ng were used to prepare sequencing libraries.

## Sequencing data processing

Sequencing reads from raw FASTQ files were trimmed and filtered using fastp. The reads were trimmed to remove adapter sequences and long homopolymer sequences (>10 base pairs). Reads with more than 50% unqualified bases (phred <15) and reads shorter than 50 base pairs for 101 bp cycles or shorter than 75 bp for 151 bp cycles were filtered from downstream analysis. After trimming and filtering, reads from Influenza A/California/07/2009 (H1N1) cultured in MDCK cells were assembled into a consensus sequence using REVICA (https://github.com/greninger-lab/revica) with the A/California/07/2009 (H1N1) isolate sequence (RefSeq GCF_001343785.1 [53]) as the initial reference. This wildtype inoculum consensus sequence was used as the reference for variant analysis in the subsequent experiments. Variant analysis was performed using RAVA (https://github.com/greninger-lab/lava/tree/Rava_Slippage-Patch). Samples with average depth <20 were excluded from the *in vitro* sequencing experiments. Samples with depth <20 at any of the lineage-associated mutation loci were excluded from the *in vivo* sequencing experiments. *In vivo* samples with low coverage by hybridization capture were re-sequenced. The reads from each replicate were analyzed separately, then combined if both replicates had similar allele frequencies at each lineage-associated locus. Raw reads are available in NCBI BioProject PRJNA1028360 (S1 to S4 Data).

## Recovery of recCA09 variants

HEK 293T cells ($2.5 \times 10^5$/ml) were seeded in 6-well plates with DMEM (7.5% FBS). After overnight incubation, wells were transfected with 0.5 μg DNA total concentration of eight (NA, M1, NP, NS, PA, PB1, PB2, HA/HA-nLuc/HA-maxGFP) viral plasmids, followed by 14-hour incubation and media replacement with DMEM containing 1 μg/ml tosylsulfonyl phenylalanyl chloromethyl keton (TPCK)-trypsin. Two days after media replacement, 2 ml of cell culture supernatant was collected and transferred to fresh MDCK cells, followed by incubation for 2 days collection of culture supernatants, amplification of recovered virus populations, and $TCID_{50}$ titration. Total RNA was extracted from the infected MDCK cell supernatants and viral genomes authenticated by Sanger sequencing and/or whole genome sequencing.

## Dose-response inhibition assays

To measure inhibition of recCA09-nanoLuc reporter viruses, 3-fold serial dilutions of 4'-FlU were prepared in 96-well plate format. MDCK cells were seeded in white-clear bottom 96 well plates ($1.5 \times 10^4$ cells/well) and 4'-FlU serial dilutions added. After incubation for 30 hours, luciferase activities in each well were determined, and raw data normalized for equally infected wells that had received vehicle (DMSO) volume equivalents. For virus yield-based dose-response assays with non-reporter viruses, MDCK cells were plated in 24 well plates ($1 \times 10^5$/well), infected with the different virus lineages as specified (MOI 0.01 $TCID_{50}$ units/cell), and compound added in 3-fold serial dilutions. After 3 days of incubation, progeny virus titers were measured by $TCID_{50}$ titration.

## Fluorescence microscopy

MDCK cells were plated in 96 well plates ($1.5 \times 10^4$/well) and incubated overnight. Following infection with an MOI of 0.01 $TCID_{50}$ units/cell and addition of 4'-FlU after removal of the virus inoculum, plates were incubated for 60 hours at 37°C. Microphotographs were taken at 10× magnification at a ZEISS Observer.D1 fluorescence microscope, using the X-Cite Series

120Q; AxioCam MRc software package (AxioVision Rel. 4.8). All fluorescence conditions were examined and documented in two independent repeats.

## Viral fitness tests

MDCK cells ($2.5 \times 10^5$/well) in 6-well plate format were infected with a mixture of mutant and genetic parent recCA09 at an MOI ratio of 0.1 (mutant) to 0.01 (WT) $TCID_{50}$ units/cell. Culture supernatants were harvested after 72-hour incubation, subjected to $TCID_{50}$ titration and total RNA extraction, and infection of fresh MDCK cells at an MOI of 0.01 $TCID_{50}$ units/cell. The procedure was repeated for a total of 5 passages and all RNA extracts subjected to whole genome sequencing.

## Viral growth curves

95%-confluent MDCK cells were infected with mutated influenza viruses at an MOI of 0.01 $TCID_{50}$ units/cell, followed by incubation in TPCK-trypsin media at 37°C and 5% $CO_2$. Cell supernatants were harvested in 12-hour intervals for a total experiment window of 84 hours and progeny virus titers determined through $TCID_{50}$ titration. Growth curves for each variant were generated in three independent experimental repeats.

## Homology modeling and structural mapping of 4'-FlU resistance mutations

Homology models of H1N1 CA09 polymerase were generated based on the coordinates reported for influenza C (PDBID 5d9a) [35] or bat influenza A (PDB 4wsb) [36] polymerase. Homology models were generated using the SWISS-MODEL homology modeling server. Spatial organization of 4'-FlU mutations was determined by mapping each mutation in Pymol. The structure of authentic 1918 H1N1 influenza polymerase (PDBID 7ni0) [34] was used to verify the accuracy of homology models. Although each structure analyzed were in pre-initiation states, the locations of all identified mutations, except Y488C and T491M, were conserved across all models. The positions of the Y488C and T491M differed between the homology model based on the bat influenza virus (PDBID 4wsb) and the other structures, reflecting a different orientation of the PB2 cap binding domain.

## Sequence alignments

Mutated residues were aligned to all complete and partial influenza A virus sequences available in the NIH NCBI Virus sequence database. FASTA files were downloaded and opened in MAFFT version 7. Sequences were aligned and alignments downloaded, opened in Jalview (2.11.2.7), and the relative frequencies of allele polymorphisms at 4'-FlU resistance sites determined.

## Intranasal infection of mice

Female mice (6–8 weeks of age, Balb/c) were purchased from Jackson Laboratories. Upon arrival, mice were rested for at least 5 days, then randomly assigned to study groups and housed under ABSL-2 conditions for infections with recCA09. Bodyweight was determined twice daily, body temperature determined once daily rectally. For infections, animals were anesthetized with isoflurane, followed by intranasal inoculation with $1 \times 100$–$5 \times 10^3$ $TCID_{50}$ units/animal of recCA09 variants. In all cases, virus inoculum was administered in amounts of 50 μl/animal (25 μl/nare). Animals were euthanized and organs harvested at predefined time points or when animals reached predefined humane endpoints.

## Pathogenicity and efficacy studies in mice

Mice were inoculated intranasally with IAV stocks as above, followed in efficacy studies by q.d. oral treatment with 4'-FlU at 2 or 10 mg/kg BW, starting 24 hours after infection, for 5 doses total. Each study contained animals receiving equal volumes of vehicle through oral gavage. At study end, lung tissues were harvested and subjected to virus titration or RNA extraction and whole genome sequencing.

## Ferret transmission studies

Female ferrets (5–8 months of age) were purchased from Triple F Farms. Upon arrival, ferrets were rested for 1 week, then randomly assigned to study groups and housed under ABSL-2 + conditions. Bodyweight and body temperature (rectally) were determined once daily. Dexmedetomidine/ketamine anesthetized animals were inoculated intranasally with $1 \times 10^5$ $TCID_{50}$ units of recCA09 in a volume of 500 μl per nare (1,000 μl per animal). Nasal lavages were performed twice daily using 1 ml of PBS containing 2× antibiotics-antimycotics (Gibco). Oral treatment with 2 mg/kg bodyweight 4'-FlU or vehicle control (source animals only) was initiated 12 hours after infection, and continued q.d. for 3 doses total. Beginning 2.5 dpi, uninfected and untreated contact animals were added to the infected and treated source ferrets in a 1:1-ratio to allow direct-contact transmission. Co-housing continued until termination of the source ferrets 4 days after the original infection. Respiratory tissue samples (lungs, tracheas, and nasal turbinates) were extracted for virus titrations and whole genome sequencing (lungs only). Sentinels were subjected to once daily nasal lavages and monitored until study day 8, when respiratory tissues were harvested and analyzed.

## Virus titration and whole genome sequencing from *in vivo* tissue samples

Organs were weighed and homogenized in 300 μl PBS (for titrations) or 300 μl RLT (for RNA extraction) using a beat blaster, set to 3 cycles of 30 seconds each at 4˚C, interspersed with 30-second rest periods. Homogenates were cleared (10 minutes at $10,000 \times g$ and 4˚C), and cleared supernatants stored at -80˚C until virus titration/ spin through a QIA shredder column (Qiagen). Viral titers were expressed as $TCID_{50}$ units per gram source tissue.

## Statistical analyses and software

For statistical analysis of studies consisting of two groups, unpaired two-tailed t-tests were applied. When comparing more than two study groups, 1-way or 2-way analysis of variance (ANOVA) with multiple comparison *post hoc* tests as specified were used to assess statistical difference between samples. The number of individual biological replicates (n values) and exact P values are shown in the figures or schematics. The threshold of statistical significance (α) was set to 0.05. Source data and statistical analysis are shown in the S5 Data and S6 Data files, respectively. Statistical analyses were carried out in Prism version 10.0.3 (GraphPad), graphs were assembled in R, Prism 10.0.3, Adobe illustrator 2022, or PyMOL. Sequence alignments were performed with MAFFT version 7 and Jalview 2.11.2.7. Cartoons and study design overviews were created in Biorender and assembled in Adobe Illustrator 2022.

## Supporting information

**S1 Table. Whole genome sequencing of adapted virus populations after passage 9 (P9) or 10 (P10) as indicated.** Shown are coding mutations that were absent in lineages passaged in the presence of vehicle (DMSO) volume equivalents. Allele frequency in parentheses; frequency cut-off 5%; read depth cut-off 50. Bold underscore denotes mutations in RdRP

subunits with >50% relative allele frequency that were rebuilt for resistance testing.
(DOCX)

**S2 Table. Dose response assays of recCA09 with rebuilt resistance mutations against 4'-FlU (EC$_{99}$ with 95% CI and fold-change EC$_{99}$ relative to parental recCA09 are shown).**
(DOCX)

**S3 Table. Dose response assays of recCA09 with rebuilt resistance mutations against favipiravir/T-705 (EC$_{99}$ with 95% confidence CI and fold-change EC$_{99}$ relative to parental recCA09 are shown).**
(DOCX)

**S4 Table. Dose response assays of recCA09 with rebuilt resistance mutations against NHC (parent compound of the prodrug molnupiravir; EC$_{99}$ with 95% confidence CI and fold-change EC$_{99}$ relative to parental recCA09 are shown).**
(DOCX)

**S5 Table. Peak virus titers and maximum growth rates of recCA09 with rebuilt resistance mutations.**
(DOCX)

**S6 Table. Representation of candidate resistance mutations in complete and partial IAV sequences available in the NIH NCBI Virus sequence database (limit: 0.001%).** Polymorphisms matching resistance mutations are shown in bold.
(DOCX)

**S7 Table. Location and predicted effect of individual resistance mutations.**
(DOCX)

**S8 Table. Recovery attempts of recCA09 with engineered combinations of independently emerged resistance mutations (genetic background: recCA09-nanoLuc).**
(DOCX)

**S1 Fig. Sterilizing dose-range finding with 4'-FlU.** Virus yield reduction assay with recCA09. Symbols represent geometric mean ± geometric SD; line shows 4-parameter variable slope regression model. EC$_{50}$ and EC$_{90}$ values are given; LoD, Limit of Detection; n = 3.
(TIF)

**S2 Fig. Repeats of microphotographs shown in (Fig 1f).** Duplicates of phase-contrast images and corresponding fluorescent images are shown for each adaptation lineage; dashed boxes denote images presented in Fig 1f; scale bar, 100 μm.
(TIF)

**S3 Fig. Dose-response assays of recCA09-nanoLuc with resistance mutations rebuilt in combination and individually. a-f)** Assessment of recCA09 with mutations found in lineages 1 (a), 2 (b), 3 (c), 4 (d), 5 (e), and 6 (f). Symbols show means ± SD, lines show 4-parameter variable slope regression models. Data were normalized for samples receiving vehicle (DMSO) volume equivalents, parental recCA09 is shown in each graph; n = 3.
(TIF)

**S4 Fig. Representation of candidate resistance mutations in complete IAV sequences available in the NIH NCBI Virus sequence database. a-c)** Shown are relative frequency of polymorphism at 4'-FlU resistance sites in PA (a), PB1 (b), and PB2 (c). Blue, predominant hydrophilic site chains; green, predominant neutral site chains; black, predominant

hydrophobic site chains. Relative size proportional to probability of presence in the database.
(TIF)

**S5 Fig. Spatial conservation of 4'-FlU resistance mutations in discrete IAV RdRP structural models. a-c)** Locations of all 4'-FlU resistance mutations in a CA09 homology model based on the coordinates released for influenza C polymerase (PDBID 5d9a) (a), the 1918 H1N1 influenza A polymerase (PDBID 7ni0) (b), and a bat influenza A polymerase (4wsb) (c). Mutations are shown as black spheres, labels are color-coded by polymerase subunit; PA, green; PB1, cyan; PB2, magenta. The active site for phosphodiester bond formation of the RdRP is shown in red. Homology models were created using SWISS-MODEL, images were created using Pymol.
(TIF)

**S6 Fig. Clinical signs in mice enrolled in pathogenesis assessment. a, c, e, g, I, k, m)** Body-weight normalized to weight at the time of infection for parental recCA09 and the rebuilt resistance lineages #1–6. Dashed line, predefined humane endpoint. **b, d, f, h, j, l, n)** Rectal temperature measured once daily. Symbols represent individual animals, lines connect data means; n = 4–5.
(TIF)

**S7 Fig. Clinical signs in mice enrolled in efficacy testing. a, c, e, g, I, k, m)** Bodyweight normalized to weight at the time of infection for parental recCA09 and the rebuilt resistance lineages #1–6. Dashed line, predefined humane endpoint. **b, d, f, h, j, l, n)** Rectal temperature measured once daily. Symbols represent individual animals, lines connect data means; n = 4.
(TIF)

**S8 Fig. Replication of 4'-FlU-resistant recCA09 lineages in mice in the presence or absence of compound. a-c)** Lung virus load 4.5 dpi of animals from (Fig 4f). Results were grouped and analyzed by treatment: vehicle (a), 2 mg/kg 4'-FlU q.d. (b), or 10 mg/kg 4'-FlU (c). Symbols represent individual animals, columns show geometric means + geometric SD. Statistical analysis with 1-way ANOVA and Dunnett's multiple comparison post-hoc test; p values are stated; n = 4.
(TIF)

**S9 Fig. Clinical signs in mice enrolled in transmission studies. a, c, e:** Bodyweight normalized to weight at the time of infection for parental recCA09 (a) and resistance lineages #4 (c) and #5 (e). Dashed line, predefined humane endpoint. **b, d, f)** Rectal temperature of recCA09 (b) and resistance lineages #4 (d) and #5 (f) measured once daily. Dashed line, onset of fever (39.5˚C). Symbols represent individual animals, lines connect data means; n = 3.
(TIF)

**S1 Data. RAVA plots covering virus adaptations.** Data can be found at https://github.com/ greninger-lab/Lieber_IAV_4FIU_Supplemental_Data_1-4/tree/main/S1_Data/adaptation_ rava_plots.
(ZIP)

**S2 Data. RAVA plots covering *in vitro* virus fitness tests.** Data can be found at https:// github.com/greninger-lab/Lieber_IAV_4FIU_Supplemental_Data_1-4/tree/main/S2_Data/ fitness_rava_plots.
(ZIP)

**S3 Data. RAVA plots covering virus populations isolated from infected mice.** Data can be found at https://github.com/greninger-lab/Lieber_IAV_4FIU_Supplemental_Data_1-4/tree/

main/S3_Data/mouse_rava_plots.
(ZIP)

**S4 Data. RAVA plots covering virus populations isolated from infected ferrets.** Data can be found at https://github.com/greninger-lab/Lieber_IAV_4FIU_Supplemental_Data_1-4/tree/main/S4_Data/ferret_rava_plots.
(ZIP)

**S5 Data. All quantitative raw data.**
(XLSX)

**S6 Data. All statistical analyses.**
(XLSX)

## Acknowledgments

We thank the Georgia State University Department of Animal Resources for assistance, M. T. Saindane for chemical synthesis, and A. I. Leach and M. R. Sirrine for help with sequence data analysis.

## Author Contributions

**Conceptualization:** Richard K. Plemper.

**Data curation:** Carolin M. Lieber, Nicole A. Lieberman.

**Formal analysis:** Alexander L. Greninger, Richard K. Plemper.

**Funding acquisition:** Richard K. Plemper.

**Investigation:** Carolin M. Lieber, Hae-Ji Kang, Megha Aggarwal, Nicole A. Lieberman, Elizabeth B. Sobolik, Jeong-Joong Yoon, Robert M. Cox.

**Methodology:** Robert M. Cox.

**Project administration:** Richard K. Plemper.

**Resources:** Michael G. Natchus.

**Supervision:** Alexander L. Greninger, Richard K. Plemper.

**Validation:** Carolin M. Lieber, Richard K. Plemper.

**Visualization:** Carolin M. Lieber, Nicole A. Lieberman, Robert M. Cox, Alexander L. Greninger.

**Writing – original draft:** Carolin M. Lieber, Richard K. Plemper.

**Writing – review & editing:** Robert M. Cox, Alexander L. Greninger, Richard K. Plemper.

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
