## [Decision Letter · Decision Letter 0]

23 Dec 2023

Dear Dr. Plemper,

Thank you very much for submitting your manuscript "Influenza A virus resistance to 4’-fluorouridine coincides with viral attenuation in vitro and in vivo" for consideration at PLOS Pathogens. As with all papers reviewed by the journal, your manuscript was reviewed by members of the editorial board and by several independent reviewers. The reviewers appreciated the attention to an important topic. Based on the reviews, we are likely to accept this manuscript for publication, providing that you modify the manuscript according to the review recommendations.

Sincerely,

Peter Palese

Academic Editor

PLOS Pathogens

Kanta Subbarao

Section Editor

PLOS Pathogens

Kasturi Haldar

Editor-in-Chief

PLOS Pathogens

orcid.org/0000-0001-5065-158X

Michael Malim

Editor-in-Chief

PLOS Pathogens

orcid.org/0000-0002-7699-2064

Reviewer Comments (if any, and for reference):

Reviewer's Responses to Questions

**Part I - Summary**

Reviewer #1: This study provides a comprehensive exploration of the resistance mechanisms to 4’-fluorouridine (4’-FlU), a small molecule previously described as a broad-spectrum RdRp inhibitor. The prediction of antiviral resistance during pre-clinical development of a novel molecule and its potential implications on viral fitness and transmission is of a major importance to the field, since resistance to antivirals has been continuously described, especially among RNA viruses. Overall, the manuscript is well-written and I think it is very suitable for Plos Pathogens. Here are some comments and suggestions for the MS:

1) Earlier studies showed that this compound has broad-spectrum activities - hence, it would be helpful to discuss and speculate its resistance profiles on other viruses.

2) The escape mutations predominantly characterized in this study were found primarily within the RdRP subunits. Have any mutations been identified in other viral RNA segments? If such mutations have been detected, please include the corresponding data in the supplementary section.

3) Is there any data on the resistance-profiling of 4’-FlU using other IAV strains and influenza B? If not, the authors may need to soft the following statements such as "these results demonstrate that partial viral escape from 4’-FlU is feasible in principle, but escape mutation clusters are unlikely to reach clinical significance or persist in circulating influenza virus strains." (lines 37-39), "these results define a high genetic barrier of 4’-FlU against influenza virus escape from inhibition." (lines 94,95), and "influenza virus escape from the developmental drug 4’-FlU is mandatorily linked to viral attenuation and reduced transmission competence." (lines 361, 362).

4) Authors could define the abbreviation 4`-FIU at first appearance in the text.

5) Line 103 - should be "grow" instead of "growth".

6) Legend of Figure S2 should be corrected: “Repeats of microphotographs shown in (Fig. 1e)” and “dashed boxes denote images presented in Fig 1e” – It should be Fig 1f instead of Fig 1e.

7) Double check and standardize the presentations of EC99 and 95% CI values in the supplementary tables.

8) What is the rationale for choosing structural models based on influenza virus C (ICV), the 1918 H1N1 IAV, and a bat IAV? Discuss any potential limitations or assumptions associated with using these models for the study.

9) In the discussion on in vivo attenuation of resistant variants, provide more insight into the potential mechanisms behind this observation. This could include exploring whether the observed attenuation is due to reduced replication efficiency, altered host interactions, or other factors.

10) Address the clinical relevance of the findings. How might the observed resistance impact the use of 4’-FlU as a potential therapeutic option?

Reviewer #2: In this manuscript by Lieber et al., they continue studies on 4’-fluorouridine (4’-FIU), which is a chain-terminating broad-spectrum nucleoside inhibitor. The focus on this study is resistance profiling using A/CA/07/2009 (H1N1) as a test system for influenza A viruses. Using a recombinant CA09 virus (CA09), they generate and then characterize six independent escape lineages. The resulting escape variants have moderate resistance to the drug. The mutations are located in proximity to the active site. Fitness of resistance variant viruses in cell culture and in mice was reported to be attenuated. Attenuation of the variants was also observed in ferrets. For example, the variants (#4 and #5) had only very low levels of virus growth in the lungs of ferrets even in the absence of the drug. Also, one of the two variants did not display contact transmission, while the other variant did. Overall, the work describes several resistance variants to the compound and demonstrates that these variants have reduced fitness in vivo. Moreover, they did detect the presence of revertants in the ferrets. This work adds to the foundation of work on the drug, encourage further development.

**Part II – Major Issues: Key Experiments Required for Acceptance**

Reviewer #1: None

Reviewer #2: (No Response)

**Part III – Minor Issues: Editorial and Data Presentation Modifications**

Reviewer #1: See Summary

Reviewer #2: 1. The recombinant CA09 virus (CA09) should be described in the Materials and Methods section under the category of Cells and Viruses.

2. Does the recombinant CA09 virus contain mouse-adaptive mutations? Or does it contain internal genes from another virus that is more lethal in mice such as PR8? I ask this because normally, A/CA/07/2009 (H1N1) is only of moderate pathogenicity in mice. However, in Figure 4 the wild-type CA09 virus they use to infect Balb/c mice is reported to have a LD50 < 1 TCID50. This is an extremely low LD50 in mice for an A/CA/07/2009 virus.

3. If the recombinant CA09 virus contains an alternative backbone or mouse-adaptive mutations, this could affect the phenotypes in ferrets. From my reading, though, it sounds like virus recovered from the resistant lineages (4 and 5) were used for the ferret experiment instead of the recombinant virus. Could the authors please clarify?

4. In Figure 4 (mouse experiment), panel h compares in each panel vehicle to increasing concentrations of the drug for each virus separately. And each graph performs a statistical analysis. Can they also compare the virus titers of the different viruses in three panels and perform statistical analyses? This would be one panel for vehicle comparing the viruses, one panel for 2 mg/kg drug, and one panel for 10 mg/kg drug.

5. On page 3, line 62, please change “99% of seasonal H1N1 isolates was” to “99% of seasonal H1N1 isolates were”. As far as I am aware, the rule is that to determine if % is a singular or plural noun, look at the noun following it. If the next noun is a plural, use a plural verb. If the next noun is singular, use a singular verb.

PLOS authors have the option to publish the peer review history of their article (what does this mean?). If published, this will include your full peer review and any attached files.

Reviewer #1: No

Reviewer #2: No

Figure Files:

Data Requirements:

Reproducibility:

References:

---

## [Decision Letter · Decision Letter 1]

22 Jan 2024

Dear Dr. Plemper,

We are pleased to inform you that your manuscript 'Influenza A virus resistance to 4’-fluorouridine coincides with viral attenuation in vitro and in vivo' has been provisionally accepted for publication in PLOS Pathogens.

Best regards,

Peter Palese

Academic Editor

PLOS Pathogens

Kanta Subbarao

Section Editor

PLOS Pathogens

Kasturi Haldar

Editor-in-Chief

PLOS Pathogens

orcid.org/0000-0001-5065-158X

Michael Malim

Editor-in-Chief

PLOS Pathogens

orcid.org/0000-0002-7699-2064

Reviewer Comments (if any, and for reference):

Reviewer's Responses to Questions

**Part I - Summary**

Reviewer #1: All the issues raised by me were well addressed. Accept.

Reviewer #2: (No Response)

**Part II – Major Issues: Key Experiments Required for Acceptance**

Reviewer #1: N/A

Reviewer #2: (No Response)

**Part III – Minor Issues: Editorial and Data Presentation Modifications**

Reviewer #1: N/A

Reviewer #2: (No Response)

PLOS authors have the option to publish the peer review history of their article (what does this mean?). If published, this will include your full peer review and any attached files.

Reviewer #1: No

Reviewer #2: No

---

## [Editor Report · Acceptance letter]

29 Jan 2024

Dear Dr. Plemper,

We are delighted to inform you that your manuscript, "Influenza A virus resistance to 4’-fluorouridine coincides with viral attenuation in vitro and in vivo," has been formally accepted for publication in PLOS Pathogens.

Best regards,

Michael Malim

Editor-in-Chief

PLOS Pathogens

orcid.org/0000-0002-7699-2064